# Genetically tunable frustration controls allostery in an intrinsically disordered transcription factor

Jing Li[1,2], Jordan T White[1], Harry Saavedra[1,2], James O Wrabl[1,2], Hesam N Motlagh[1,2], Kaixian Liu[1], James Sowers[1], Trina A Schroer[1], E Brad Thompson[1,3], Vincent J Hilser[1,2]*

[1]Department of Biology, Johns Hopkins University, Baltimore, United States; [2]TC Jenkins Department of Biophysics, Johns Hopkins University, Baltimore, United States; [3]Center for Nuclear Receptors and Cell Signaling, Department of Biology and Biochemistry, University of Houston, Houston, United States

**Abstract** Intrinsically disordered proteins (IDPs) present a functional paradox because they lack stable tertiary structure, but nonetheless play a central role in signaling, utilizing a process known as allostery. Historically, allostery in structured proteins has been interpreted in terms of propagated structural changes that are induced by effector binding. Thus, it is not clear how IDPs, lacking such well-defined structures, can allosterically affect function. Here, we show a mechanism by which an IDP can allosterically control function by simultaneously tuning transcriptional activation and repression, using a novel strategy that relies on the principle of 'energetic frustration'. We demonstrate that human glucocorticoid receptor tunes this signaling in vivo by producing translational isoforms differing only in the length of the disordered region, which modulates the degree of frustration. We expect this frustration-based model of allostery will prove to be generally important in explaining signaling in other IDPs.
DOI: https://doi.org/10.7554/eLife.30688.001

*For correspondence:
hilser@jhu.edu

## Introduction

A cornerstone of biological regulation is the ability of proteins to tune their particular activities in response to the binding of specific ligands at distinct regulatory sites (*Motlagh et al., 2014*). Historically, such tunability has been explained by the concerted (*Monod et al., 1965*) or sequential (*Koshland et al., 1966*) models of allosteric regulation, which describe the coupling between binding sites in terms of ligand-induced changes in the average structure of the protein. More recent studies reveal that allostery is not restricted to structured proteins. It is widely observed in intrinsically disordered (ID) proteins, polypeptides, or regions therein, that lack stable tertiary structure (*Ferreon et al., 2013*; *Garcia-Pino et al., 2010*; *Lum et al., 2012*; *Motlagh et al., 2014*; *Sevcsik et al., 2011*). Moreover, ID regions are hyper-abundant in known allosteric proteins such as transcription factors (*Gronemeyer and Bourguet, 2009*; *Liu et al., 2006*), suggesting that allostery involving ID sequences may represent a major regulatory paradigm. Despite the existing evidence, however, the mechanism by which ID proteins facilitate allostery is not known.

Previously, we developed a mathematical model to show how proteins could use intrinsic disorder to facilitate, and even optimize, allosteric control (*Hilser and Thompson, 2007*). This model predicts that coupled folding and binding in different ID domains could produce complex coupling mechanisms that result from the simultaneous tuning of both activating and repressing sub-ensembles within the overall conformational ensemble (*Hilser et al., 2006, 2012*; *Motlagh et al., 2014*), a process of 'energetic frustration' akin to the well-known physical concept of 'geometric frustration'. In

**eLife digest** Proteins carry out most of the key tasks inside cells. To perform these roles, proteins must fold up to form complex three-dimensional structures. Researchers used to think that the useful parts of proteins all had set structures. However, we now know that 'disordered' proteins with variable structures are common and disordered parts of proteins can have vital roles.

In a process called allosteric regulation, regulator molecules can increase or decrease the activity of a protein by binding to it. This binding was thought to work by changing the structure of the protein, but it was not clear how this works in disordered proteins. To investigate, Li et al. studied a disordered protein called glucocorticoid receptor, and found that disordered regions can have opposing effects on other regions of the protein. This creates a 'tug-of-war' that Li et al. term "energetic frustration", whereby the activity of the protein results from the combination of the opposing interactions.

Further investigation revealed that the glucorticoid receptor produces different versions of itself that have different degrees of energetic frustration, which alters how effectively the proteins perform their tasks. This means that the protein can regulate its own activity even in the absence of binding to regulator molecules.

The concept of energetic frustration could enhance our understanding of the many different proteins that contain disordered regions. Eventually, this knowledge could be used to develop drugs that alter the activity of these proteins and so could form part of treatments for a wide range of conditions including autoimmune diseases (such as rheumatoid arthritis and lupus), cancers, and organ rejection for transplant patients. The results presented by Li et al. suggest where more research is needed to achieve this goal. For example, we need to understand more about the stability of disordered protein regions, and to identify which surfaces of the proteins interact with each other.

DOI: https://doi.org/10.7554/eLife.30688.002

condensed matter physics, 'geometric frustration' describes a physical system's inability to simultaneously minimize the competing interaction energies between its components in mean field theory (*Vannimenus and Toulouse, 1977*; *Villain, 1977*). Frustration theory has been invaluable in understanding magnetic and superconducting systems (*Vannimenus and Toulouse, 1977*), circuits (*Wang et al., 2006*), protein folding (*Bryngelson and Wolynes, 1987*), and even gene networks (*Krishna et al., 2009*). However, whereas numerous biological networks can utilize multiple components (e.g. repressors and activators) to control overall activity, it is not known whether a single gene product could encode tunable activity based on an analogous form of frustration, as theory predicts (*Hilser et al., 2006*; *Hilser et al., 2012*; *Motlagh et al., 2014*).

To investigate the relationship between disorder and allostery and to test whether energetic frustration is at the heart of disorder-mediated allostery, we selected the human glucocorticoid receptor (GR) as a model system. The GR is a member of the steroid hormone receptor (SHR) family of transcription factors and plays key roles in organ development, metabolite homeostasis, and the responses to stress and inflammation (*Griekspoor et al., 2007*). Three major domains segregate the GR's primary functions (*Hilser and Thompson, 2011*). The DNA-binding domain (DBD) and ligand-binding domain (LBD) are well-structured and are responsible for interacting with DNA (i.e. GR response element) and the steroid hormone (e.g. cortisol), respectively (*Hilser and Thompson, 2011*). The N-terminal domain (NTD), which consists of the first 420 amino acids, contains the activation function 1 core region (i.e. AF1c, GR 187–244), which is required for the recruitment of cofactors necessary for transcriptional activation (*Dahlman-Wright et al., 1994*; *Ford et al., 1997*) and full transcriptional potency. In contrast to the LBD and DBD, the NTD of GR is intrinsically disordered (*Hilser and Thompson, 2011*). Importantly, five active isoforms (*Figure 1*, Inset) among the total of eight translational isoforms of GR (*Figure 1—figure supplement 1*), differing only in the lengths of the ID NTDs have been discovered (*Lu and Cidlowski, 2005*). These isoforms differ in their relative activities (*Bender et al., 2013*), tissue distributions (*Lu and Cidlowski, 2005*; *Lu and Cidlowski, 2006*), and regulatory specificities (*Bender et al., 2013*; *Cao et al., 2013*; *Lu and Cidlowski, 2005*; *Lu and Cidlowski, 2006*). Although the effect of binding either different steroid molecules

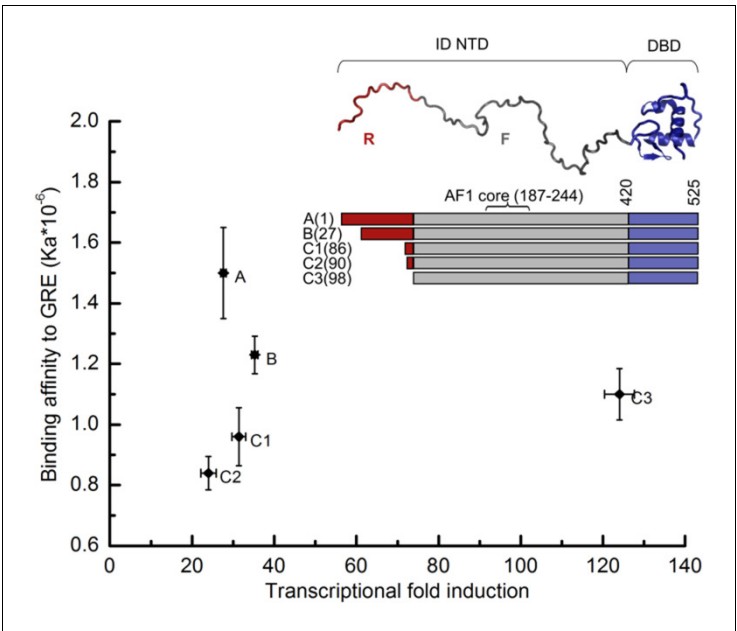

**Figure 1.** Transcriptional activity and DNA-binding affinity are not correlated among GR translational isoforms. Transcriptional activity, monitored by dual luciferase reporter assay (*Figure 1—figure supplement 1a and b''*), and binding affinity to DNA, monitored by fluorescence anisotropy change (*Figure 1—figure supplement 1c and d*), show uncorrelated behavior. **Inset**: Domain organization of the constitutively active GR constructs for translational isoforms, wherein the intrinsically disordered N terminal domain (NTD) varies in length. Residues 1–97 (red) are labeled R (for Regulatory) and residues 98–420 (grey) are labeled F (for Functional). Also labeled are residues corresponding to the activation function 1 core (AF1 core) region, which is required for transcriptional activity (*Ford et al., 1997*). Error bars represent uncertainty of the individual fits.
DOI: https://doi.org/10.7554/eLife.30688.003

The following figure supplement is available for figure 1:

**Figure supplement 1.** Transcriptional activity and DNA-binding affinity of GR translational isoforms.
DOI: https://doi.org/10.7554/eLife.30688.004

(*Pandit et al., 2002*; *Pfaff and Fletterick, 2010*) or different DNA sequences (*Meijsing et al., 2009*) is known to produce a variety of structural changes within their respective binding domains, how such binding events are differentially propagated to the ID NTD of each isoform, and subsequently translated into functional changes, is not known. Fundamentally, it is not clear whether and, if so, how structured domains like the DBD can both receive and transmit allosteric signals to disordered domains like the NTD of GR.

## Results and discussion

### Thermodynamic coupling underlies tunable DNA-binding affinity and transcriptional activity of different isoforms

To obtain insight into how allostery tunes GR function, DNA binding and cell-based functional studies were performed on constitutively active (i.e. steroid-independent) versions (*Chen et al., 1997*) of human GR translational isoforms that lack the C-terminal LBD (*Figure 1*). The similarity between the relative transcriptional activities of the different two-domain isoforms studied here and those of the full-length three-domain isoforms studied previously (*Bender et al., 2013*; *Lu and Cidlowski, 2005*) (*Figure 1* and *Figure 1—figure supplement 1a and b''*), suggests that although the LBD affects the magnitude of activity enhancement (*Godowski et al., 1987*; *Hollenberg and Evans, 1988*), it does not appear to qualitatively impact the communication between the DBD and the NTD in each isoform.

Several features of the activities and DNA-binding properties of the different isoforms are note-worthy. First, the affinities of the isoforms for DNA vary, despite having identical DBDs, indicating that the NTD of each isoform differentially communicates with its respective DBD (*Figure 1* and *Figure 1—figure supplement 1c–e*). Second, the GR C3 isoform (i.e. 98–525) is almost five times more active than the full-length GR A isoform (i.e. 1–525), indicating that residues 1–97 somehow negatively regulate the activity of the remaining NTD residues, which contain the functionally important AF1c region (*Figure 1*). For this reason, we represent the full-length NTD as being composed of two distinct domains, a functional domain (F-domain), and a regulatory domain (R-domain) (*Figure 1*, Inset), which were experimentally shown to be unfavorably coupled to each other by both osmolyte (i.e. TMAO) induced folding and protease sensitivity analyses (*Figure 2a and b*) (*Li et al., 2012*). Conversely, in the same in vitro folding and protease sensitivity experiments the DBD appears to stabilize the folded form of the F-domain (*Figure 2a and b*). Importantly, the thermodynamic stabilization of the F-domain conferred by the DBD is accompanied by a dramatic increase in activity, as

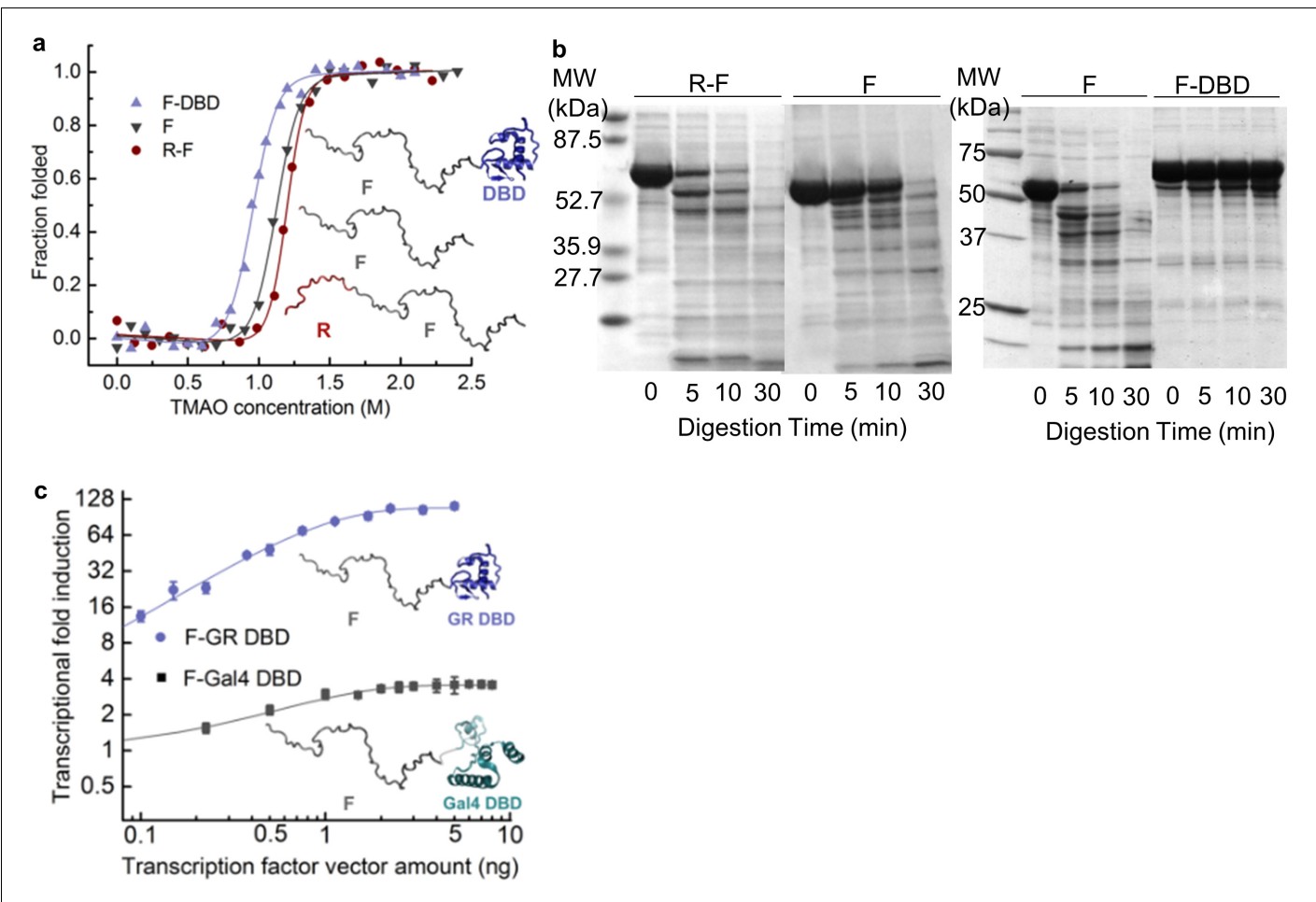

**Figure 2.** Coupling of the R-domain and DBD to the functional F-domain in GR. (a) TMAO-induced folding for the F-domain alone and with either the R-domain or the DBD. (b) Protease sensitivity assay: comparing F-domain and F-domain with R-domain (left) performed at a protein (1 mg/ml): trypsin mass ratio of 1000:1 (*Li et al., 2012*); comparing F-domain and F-domain with DBD (right) performed at a protein (1 mg/ml): trypsin mass ratio of 100:1. Note: each protein pair (i.e. F vs. R-F and F vs. F-DBD) was run on the same gel, with the intervening lanes removed in the figure for clarity. (c) Luciferase assay for C3 isoform (GR F-DBD) versus chimeric construct (GR F-Gal4 DBD). We note that data for the R-F and F domains in panels (a) and (b) are the same as presented in *Li et al., 2012* to allow for direct comparison of the opposing effects of the DBD and the R-domain on the F-domain.

DOI: https://doi.org/10.7554/eLife.30688.005

The following figure supplement is available for figure 2:

**Figure supplement 1.** Domain stabilities determined by TMAO-induced protein folding transitions.
DOI: https://doi.org/10.7554/eLife.30688.006

determined from cell-based transcriptional assays that compare the activity of the GR F-DBD construct (C3 isoform) with the non-natural chimera that tethers the DBD from the yeast Gal4 transcription factor to the F-domain (*Figure 2c*). The thermodynamic and activity results suggest that competing factors within GR determine the overall stability (i.e. the ΔG of folding) and transcriptional activity of the AF1c region of the F-domain.

The coupling between the R-domain and the DBD was further evaluated by use of competitive transfection assays to estimate the DNA-binding affinity of various constructs that connect the DBD and the R-domain of each isoform. In order to only measure the coupling between the R-domain and the DBD, and to avoid the convoluting effects associated with the coupling between the F-domain and the R-domain and DBD, constructs were generated that utilized a series of flexible linkers connecting the R-domain to the DBD (*Figure 3a*). As *Figure 3b* reveals, inclusion of the R-domain residues 1–85, naturally present in GR's A and B isoforms, significantly increased the DNA-binding affinity of the DBD, while the shorter length R-domains in the C isoforms show no effect. Such increases were not observed for non-natural chimeric constructs that linked the various R-domains to the yeast transcription factor Gal4 DBD (*Kraulis et al., 1992*) (*Figure 3b*). In addition, the fact that different length linkers (*Figure 3c*) give similar results indicates that the linker is, as intended, functionally inert, and serving as a tether that simply connects the R-domain to the DBD. These results support the notion that the R-domain affects DNA binding by the DBD specifically through stabilization of a high-affinity state of the DBD, and is not simply a consequence of a direct interaction between the R-domain (or the linker) and the DNA. In other words, our results indicate the R-domain allosterically affects DNA binding of the DBD, serving as a positive intramolecular allosteric effector.

## Energetic frustration in GR

The analysis of the thermodynamic couplings in *Figures 2* and *3* point to the paradoxical result whereby the binding of DNA to the DBD simultaneously produces two opposing effects on the F-domain. It has long been known that DNA binding to GR DBD stabilizes the DBD (*Lefstin and Yamamoto, 1998*). As a consequence of its direct positive coupling to the F-domain (*Figure 2*), DBD binding to DNA stabilizes the folded form of the F-domain and therefore promotes activation of transcription (*Figure 4a*, counterclockwise green arrow). However, as *Figure 3* indicates, there is also positive coupling between the DBD and the R-domain. But because of the negative coupling between the R- and F-domains (*Figure 2a and b*), the same DNA binding (to the DBD) that stabilizes/activates the F-domain simultaneously, through stabilization of the R-domain, destabilizes the folded form of the F-domain, promoting repression of transcription (*Figure 4a*, red inhibitory semicircle), a process that resembles geometric frustration (*Vannimenus and Toulouse, 1977*; *Villain, 1977*).

Geometric frustration originates in bi-stable systems wherein competing thermodynamic couplings interact such that no single state acquires significant stability so as to dominate the ensemble probabilities (*Krishna et al., 2009*; *Vannimenus and Toulouse, 1977*; *Villain, 1977*). Within the context of the three domains of GR studied here, eight different configurations of coupling energies (i.e. *Figure 4bi–viii*) represent all possible combinations of positive (+) and negative (−) coupling energies between domains. For each case, a positive interaction energy signifies that a stabilization of one domain would result in a stabilization of the second domain, whereas negative coupling would produce the opposite effect. As *Figure 4b* reveals, frustration results when the 'direct' impact of stabilization of the DBD on the F-domain is opposite in sign to the indirect impact (i.e. the impact that is mediated through the R-domain). Such is the case when one or all three of the inter-domain interactions is/are negative (*Hilser et al., 2012*; *Motlagh and Hilser, 2012*; *Motlagh et al., 2014*). Of the possible configurations that are predicted to produce frustration (*Figure 4b* upper), GR clearly conforms to case ii, wherein the DBD is positively coupled to the F-domain serving to increase its activity (*Figure 2a–c*). However, because of the negative coupling between the R-domain and the F-domain (*Figure 2a and b*), and the positive coupling between the R-domain and the DBD (*Figure 3*), the DBD is ultimately also negatively coupled to the F-domain. The net effect of DNA binding on GR transcriptional activity is thus a balance between the strengths of these coupling energies, which could differ among translational isoforms. The results clearly demonstrate that two opposing control mechanisms are at play, and that the classic deterministic models of allostery

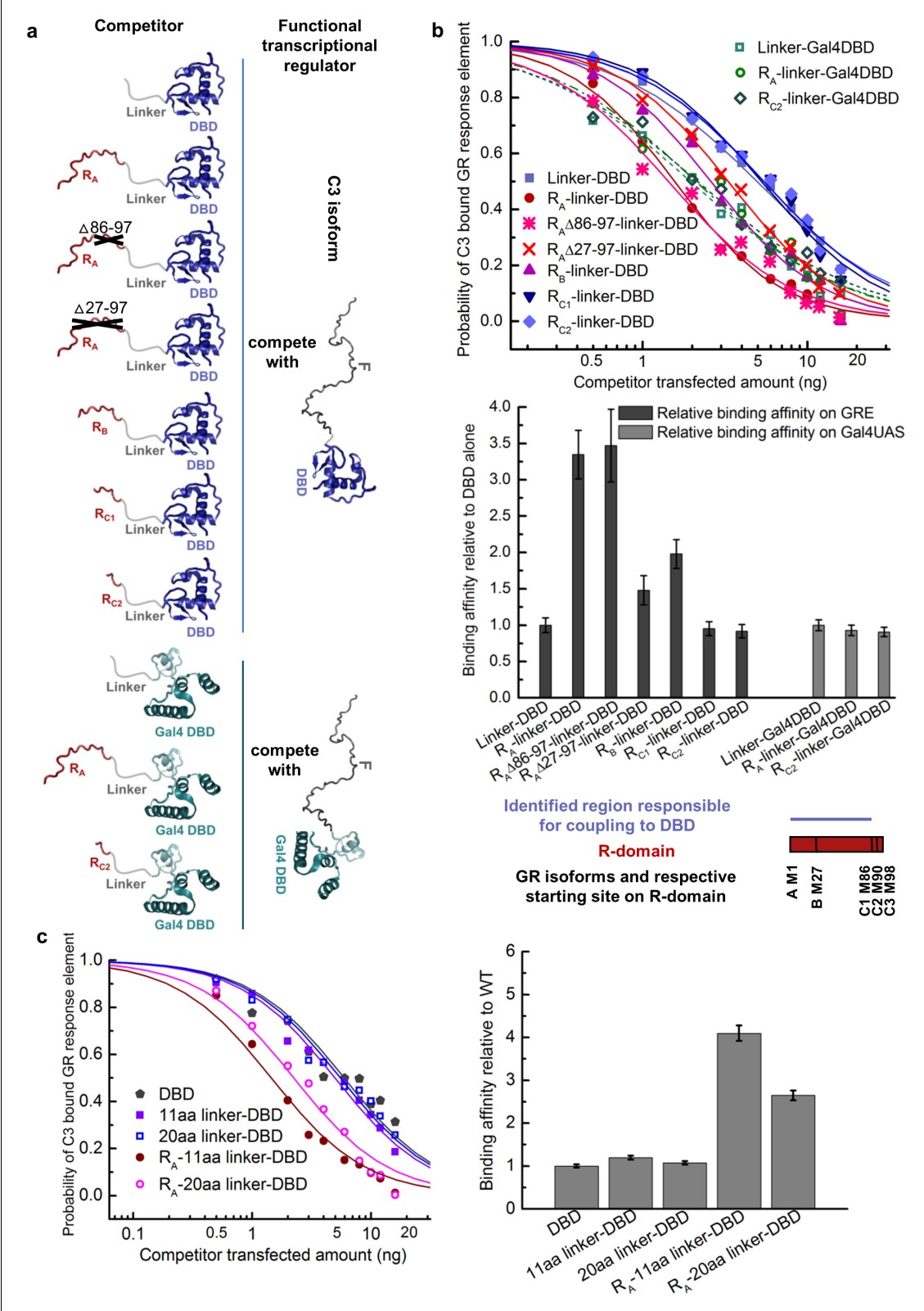

**Figure 3.** Coupling between the R-domain and DBD in GR. (a) Schematic representation of competitive transfection assay design for constructs of the R-domains of isoforms A, B, C1, and C2 as well as the internally truncated R-domain linked to GR DBD or Gal4 DBD. (b) Experimental competitive transfection assay for constructs shown in a. The first 86 residues of R-domain present in GR A and B isoforms significantly increase the binding affinity of the DBD to GRE. Linking the R-domains to the Gal4 DBD ablate coupling. (c) Linkers of 11aa and 20aa were compared for the effect when

*Figure 3 continued on next page*

*Figure 3 continued*

connected alone to the GR DBD or used to link the R-domain (GR 1–97 segment) to the DBD. Linker length in panel a and all the other figures is 11aa. Error bars represent uncertainty of the individual fits.

DOI: https://doi.org/10.7554/eLife.30688.007

(*Koshland et al., 1966*; *Monod et al., 1965*) are insufficient to capture the probabilistic nature of this mechanism.

To highlight the analogy with geometric frustration, while simultaneously distinguishing this biological phenomenon from the condensed matter physics model, we term this phenomenon 'energetic frustration'. The simplest models for geometric frustration quantify the total energy of a system of spins within a magnet, where the energy of an interacting spin pair of nuclei $i$ and $j$ takes the form $E_{int} = J_{ij}S_iS_j$. In this formalism, $J$ is the coupling energy and $S$ is the spin state (i.e. 'up +1' or 'down −1'). The interaction energy between protein domains $i$ and $j$ can be written in a similar way, $E_{int} = J_{ij}S^*_iS^*_j$, with an interdomain coupling energy and an accounting for the states of the domains. One key difference, with respect to geometric frustration, is that in energetic frustration the sign of $E_{int}$ is determined solely by the sign of $J$, which is fixed by the physicochemical nature of the interdomain interaction. Also, the domain state is conditional on whether the protein domain is folded or unfolded, a folded state resulting in a + 1 value for $S^*$ and an unfolded state resulting in a value of 0. Only in the case in which both domains are folded will the coupling energy contribute to the system.

## The ensemble allosteric model (EAM) enables quantitative characterization of the energetic frustration in GR

To illuminate how the opposing allosteric mechanisms are manifested in our example IDP, the GR, a quantitative characterization of the allosteric coupling was implemented using the previously developed ensemble allosteric model (EAM) (*Hilser and Thompson, 2007*; *Hilser et al., 2012*). An ensemble of states was constructed for each isoform, enumerating all possible combinations of the DBD being in the high-affinity or low-affinity states, and the R and F-domains being in their active (folded) or inactive (unfolded) states (see *Figure 5a*, for the A and C3 isoforms). Our choice of model is justified because as shown previously (*Li et al., 2012*) the R- and F-domains can fold to globular protein-like structures (*Figure 2—figure supplement 1b*), consistent with the notion of coupled folding and binding for IDPs, where the folded conformation is the active form (*Dyson and Wright, 2002*). In the context of the EAM, the probability of any state is determined by the intrinsic stability of each domain, $\Delta G_R$, $\Delta G_F$ and $\Delta G_D$ (the stability of each domain as it would be in isolation), as well as the coupling free energies between each domain, $\Delta g_{R-F}$, $\Delta g_{R-D}$ and $\Delta g_{F-D}$ (*Hilser and Thompson, 2007*; *Hilser et al., 2012*). For example, in the full length A isoform (composed of the R-domain, the F-domain, and the DBD) and the most active C3 isoform (composed only of the F-domain and the DBD), the EAM produces 8 and 4 states in their respective ensembles, representing all combinations (*Figure 5a*). In the EAM, the experimentally observed transcriptional activity is represented by the summed probabilities of states whose folded F-domain co-occurs with the high-affinity DBD conformation. Similarly, DNA-binding affinity is represented by the summed probabilities of states wherein the DBD is in the high-affinity conformation.

Using measurements of transcriptional activities and binding affinities of the five isoforms (*Figure 1* and *Figure 1—figure supplement 1a–d*), measurements of relative binding affinities of $R_A$-linker-DBD and Linker-DBD constructs (*Figure 3b*), and measurements of conformational stabilities (*Figure 2a* and *Figure 2—figure supplement 1a*) as constraints, quantitative estimates of the stabilities and coupling energies for each domain were obtained through unbiased comprehensive searches of parameter space (*Figure 5—figure supplement 1a*). Importantly, the maximum likelihood parameters (shown in *Figure 5—figure supplement 1e*) faithfully reproduce both the relative affinities and transcriptional activities for all five isoforms (*Figure 5b*). The search results demonstrated clear maximum likelihoods for each thermodynamic parameter (*Figure 5—figure supplement 1e*), indicative of the qualitative correctness of the activating and repressing scheme outlined in *Figure 4a*. In particular, in all cases, the signs of the coupling energies between domains are preserved, that is, both $\Delta g_{R-D}$ and $\Delta g_{F-D}$ are positive while $\Delta g_{R-F}$ is negative (*Figure 5—figure*

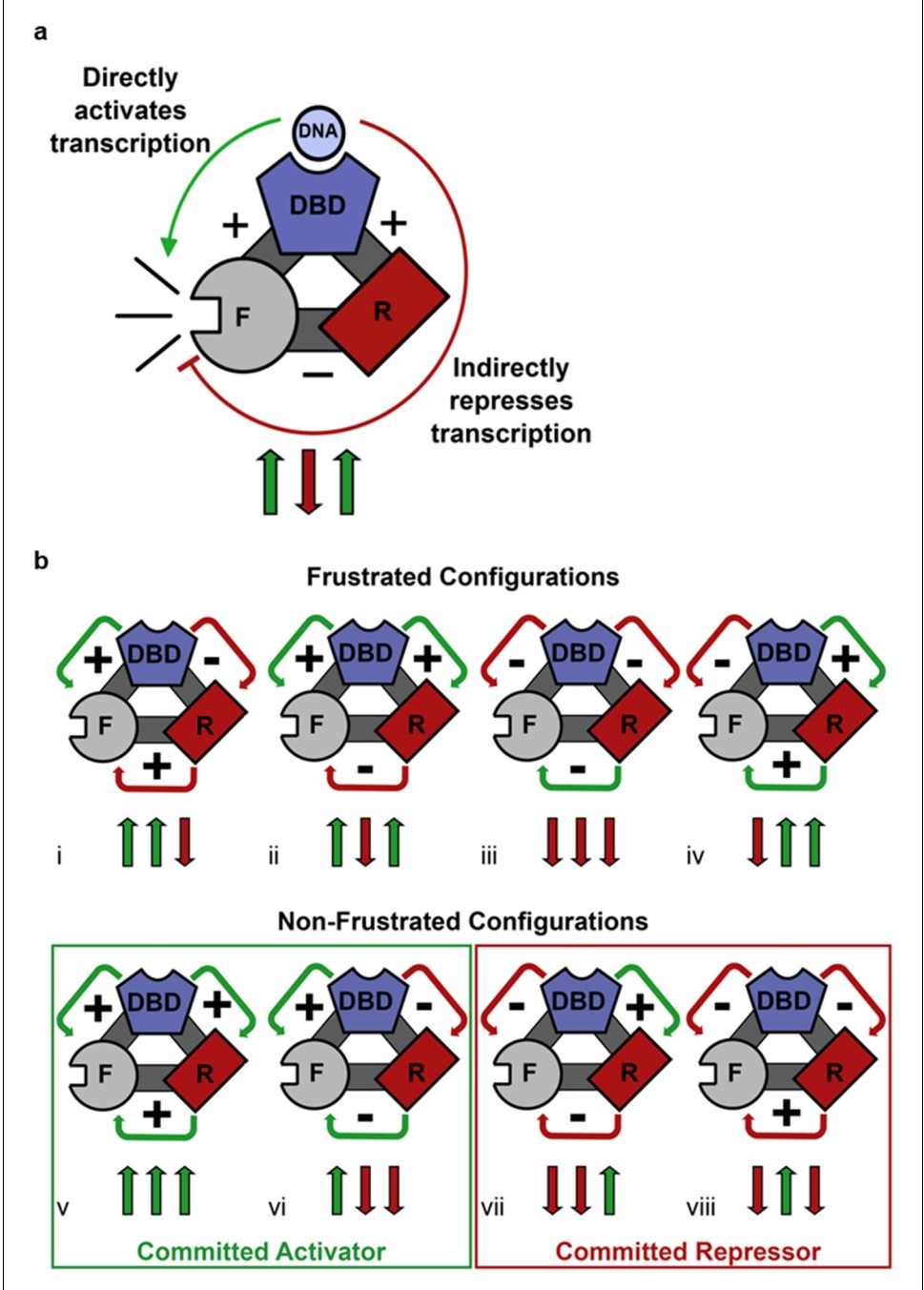

**Figure 4.** Competing thermodynamic coupling in GR produces frustration. (a) Schematic view of the thermodynamic configuration of GR. According to this convention, the positive (+) signs between the DBD and F-domain, and the DBD and R-domain signify they are positively coupled; stabilization of one domain stabilizes the other. The negative (-) sign between the R- and the F- domains indicate they are negatively coupled; stabilization of one domain destabilizes the other. Thus, the combination of the positive coupling between DBD and R-domain, and the negative coupling between R- and F-domains produces an overall repression (red semicircle) from the effect that is mediated through the R-domain. Triplet arrows are an abbreviation to describe all the couplings in the system: first arrow indicates the coupling between DBD and F-domain, second arrow indicates coupling between R-domain and F-domain, and third arrow indicates coupling between DBD and R-domain. Green (up) arrow indicates positive coupling and red (down) arrow indicates negative coupling. Importantly for this equilibrium scheme, any permutation containing an even number of up arrows will be energetically frustrated. (b) Schematic representation of all possible thermodynamic configurations for three

*Figure 4 continued on next page*

*Figure 4 continued*

interacting domains (represented in the structural and spin (up or down arrows) nomenclatures). Four of the possible configurations produce frustration between the DBD and the F-domain (top), whereas four configurations do not produce frustration (bottom), representing the cases of activation and repression, respectively. Note the curved arrows connecting the domains are colored according to the overall effect on F-domain from stabilizing the DBD; this coloring scheme highlights the result that frustrated configurations result from opposing energetic couplings affecting the F-domain stability. For example, in case i (Top, Left), stabilization of the DBD destabilizes the R-domain due to the negative coupling energy, signified by the red curved arrow between domains. However, the destabilization of the R- domain has the effect of also destabilizing the F-domain, because the F- and R- domains are positively coupled. Thus, the overall effect of stabilizing the DBD is to destabilize the F-domain through its interaction with the R-domain, signified by the red curved arrow between the R- and F- domains (even though the coupling between R-F alone is positive).

DOI: https://doi.org/10.7554/eLife.30688.008

supplement 1e), demonstrating the robustness and validity of the opposing frustration-based control mechanism underlying allostery in GR.

## Model prediction and validation

To further test the model, we sought to identify and ablate the repression component of the mechanism and quantitatively evaluated the impact on the natural GR isoforms. To do this, we targeted the interaction between the R-domain of the A isoform (i.e. $R_A$, residues 1–97) and the DBD (*Figure 1* inset), by determining the impact of point mutations in the DBD on DNA binding affinity in the presence and absence of the tethered R-domain. For constructs containing only the DBD and the R-domain, DBD mutations could perturb either the stability of the DBD, the coupling between the DBD and the R-domain, or both. Thus, by expressing the mutant forms as linker-DBD and $R_A$-linker-DBD constructs, using an inert linker to substitute for the F-domain (*Figure 6a*), the impact of the mutations could be clearly discerned. Mutations that affect stability of the DBD are predicted to affect the DNA-binding affinity of both constructs (*Figure 6b*, left). However, mutations that affect the coupling between the R-domain and the DBD are predicted to impact the DNA-binding affinity of only the $R_A$-linker-DBD construct, leaving the activity of the linker-DBD construct unaffected (*Figure 6b*, right). Screening of conserved surface residues on the DBD (*Figure 6—figure supplement 1a*), which did not significantly affect DBD stability or the coupling between the F-domain and DBD (*Figure 6—figure supplement 1b*), revealed only three positions (i.e. C431, V435, and L436) that exhibited the expected signature of DBD residues that mediate coupling to the R-domain (*Figure 6c* and *Figure 6—figure supplement 1c and d*). Consistent with these positions exerting their effects through a common mechanism, all three residues mapped to a conserved contiguous surface on the DBD (*Figure 6d*).

To qualitatively and quantitatively test the frustration-based control mechanism, the triple mutation (C431Y/V435A/L436A) was introduced into the full-length GR A isoform. Because the mutation should decrease the stabilizing effect of binding DNA on the R-domain, which in turn, should lower the destabilizing effect on the F-domain, the model predicts the counter-intuitive result whereby the activity of the triple mutant should increase, while the DBD-DNA binding affinity should decrease (*Figure 6e* Model). Importantly, such a prediction is the direct result of the frustration in the system and would represent a compelling argument for the competing energetic couplings shown in *Figure 4a*.

True to the prediction (*Figure 6e* Experiment), the effect of the triple mutant is a decrease in affinity for DNA while concomitantly increasing the transcriptional activity. This result is particularly important because while the modulatory role of the residues we refer to as the R-domain has been known (*Bender et al., 2013*), previous interpretations attributed the effect to simple steric occlusion (*Bender et al., 2013*). Our results unequivocally demonstrate that the R-domain not only negatively affects the F-domain (*Figure 2a and b*) but also positively affects the wild-type DBD (*Figure 3*), increasing the affinity of the DBD for DNA (*Figure 3*), and it does so in a manner that is directly related to the stabilities and coupling energies in the system.

Furthermore, the facts that the three residues implicated in the coupling were independently identified through mutational analysis (*Figure 6c* and *Figure 6—figure supplement 1b–d*), but

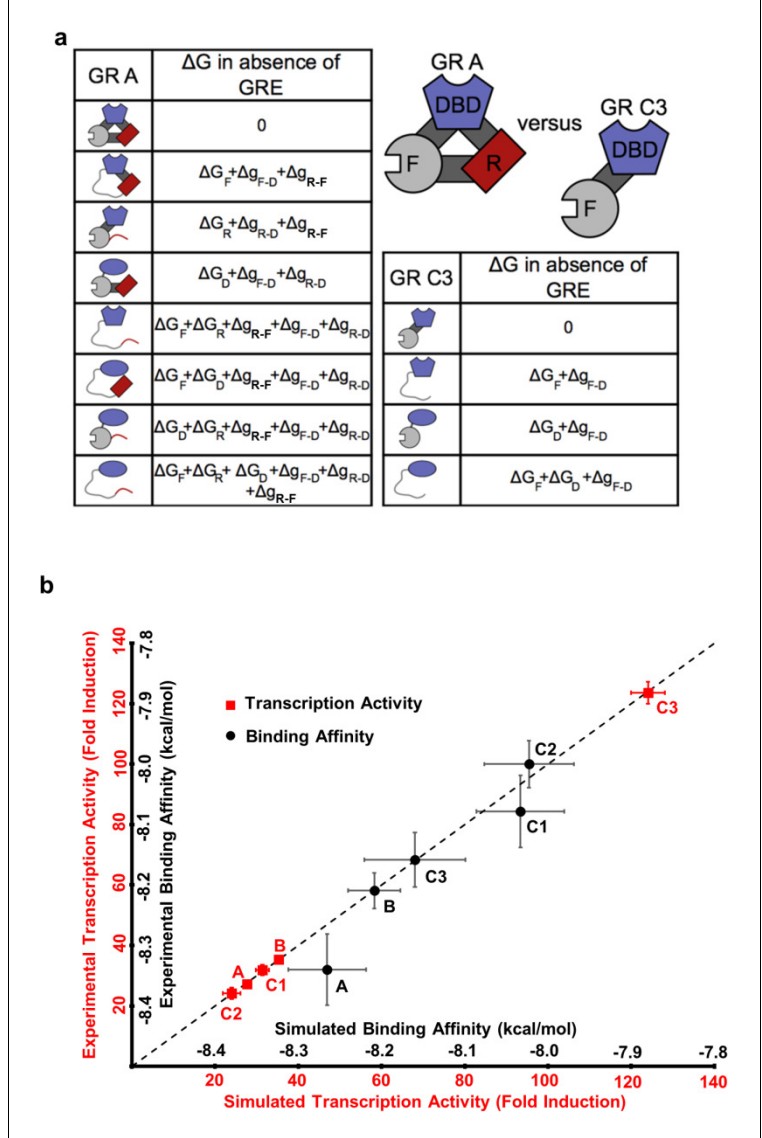

**Figure 5.** The ensemble allosteric model (EAM) quantitatively describes GR transcriptional activation. (**a**) EAM for GR A and C3 isoforms. (**b**) The model recapitulates the DNA-binding affinity and relative transcriptional activity of A, B, C1, C2, and C3 isoforms. Error bars on experimental data represent uncertainty of the individual fits. Error bars on simulated data are average results from propagation of experimental error through the model.

DOI: https://doi.org/10.7554/eLife.30688.009

The following figure supplement is available for figure 5:

**Figure supplement 1.** Probability distribution of each thermodynamic parameter from EAM.

DOI: https://doi.org/10.7554/eLife.30688.010

nonetheless mapped to a conserved contiguous surface on the DBD (*Figure 6d*), strongly supports a model whereby these residues affect the coupling between the DBD and the R-domain through a common mechanism involving direct interactions between domains. Further supporting this notion, titration of the DBD of the C3 isoform and the C3 triple mutant (C431Y&V435A&L436A) with the R-domain (expressed in trans), using the luciferase assay as a reporter, clearly shows a greater concentration dependence of activity for the wild-type C3 isoform over the triple mutant (*Figure 6f*). This result indicates that the R-domain can exert its stabilization effect on the DBD through mass action, suggestive of a direct interaction involving residues C431, V435, and L436. We note that although the combination of comparatively weak coupling energies (based on the maximum

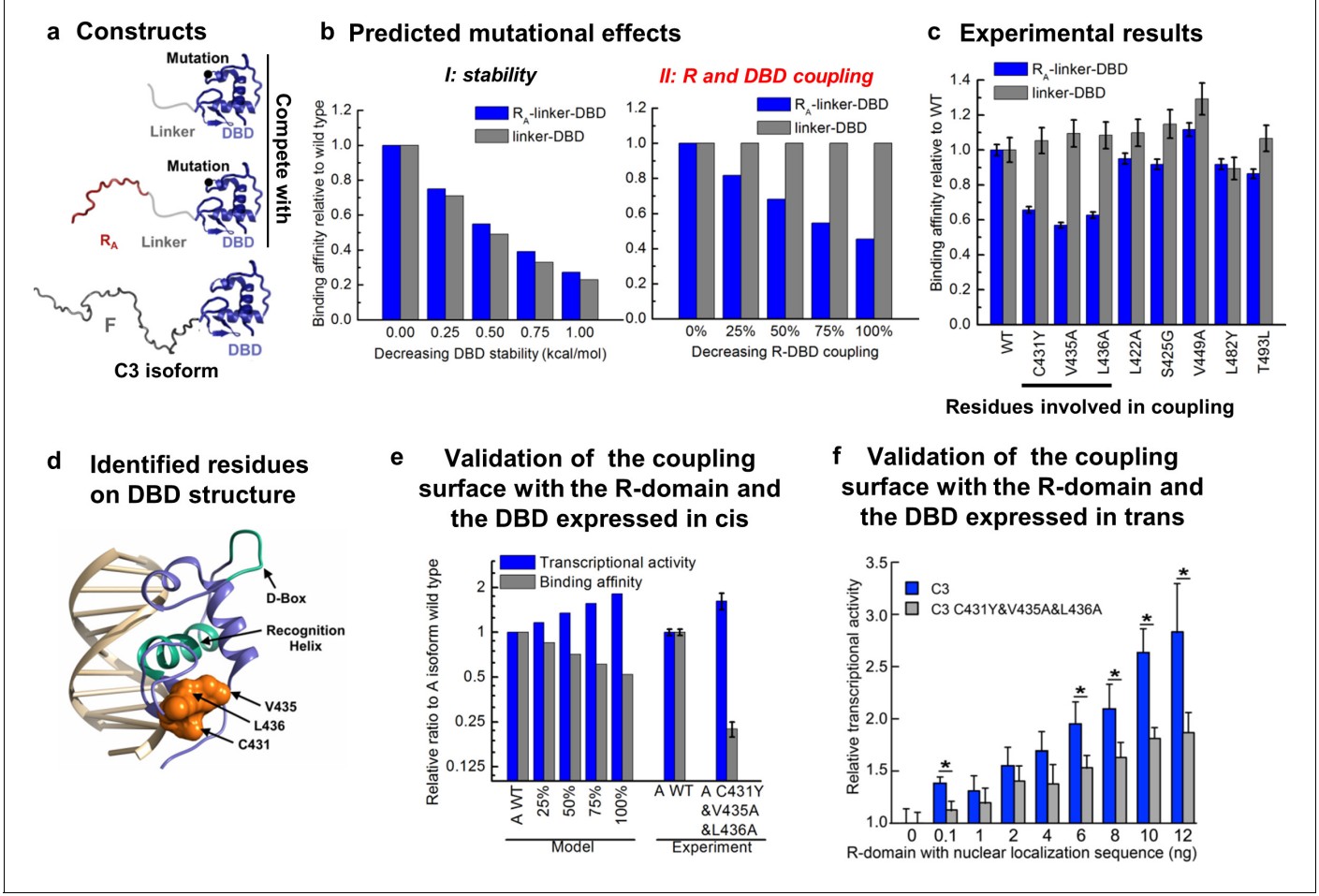

**Figure 6.** Contiguous surface mediates coupling between the DBD and the R-domain. (a) Schematic representation of the constructs used to identify residues on DBD involved in coupling to the R-domain. (b) EAM predicted changes in binding affinity for mutations that affect the stability of the DBD (left) or the coupling of the R-domain and the DBD (right) in the different constructs shown in a. (c) Experimental competitive transfection assays for single point mutants of both constructs shown in a. Error bars reflect uncertainty of the individual fits. (d) Cartoon of the proposed DBD surface (orange) involved in the coupling interaction with the R-domain. Shown also are the known DNA recognition helix (green helix) and the known dimerization interface (green loop). (e) Independent validation of the proposed coupling surface using the wild type and triple mutant of the A isoform, where the R-domain and the DBD are expressed in cis. The EAM predicted changes in transcriptional activity and DNA-binding affinity for mutations influencing the R-domain and the DBD coupling (from 25% to 100%) on A isoform (Model, left set of bars). Influence of the triple mutations (C431Y&V435A&L436A) on the transcriptional activity and the binding affinity obtained from the luciferase dosage curve of the A isoform (Experiment, right bars, detailed in *Figure 6—figure supplement 1e and f*). Error bars reflect uncertainty of the individual fits. (f) Independent validation of the proposed coupling surface with the wild type and the triple mutant of the C3 isoform, with the R-domain expressed in trans. Dual luciferase assay showing the wild type and the triple mutant (i.e. C431Y/V435A/L436A) versions of the GR C3 isoform titrated with R-domain, expressed with a nuclear localization sequence and Flag tag. Error bars are 95% confidence intervals and the asterisks indicate differences that are significantly different (p<0.01) by a T-test with the Benjamini-Hochberg correction for multiple tests. The y-axis is normalized to the initial, 0 ng point for each dataset. The EAM predictions in Panels b and e utilize parameters described in *Figure 5—figure supplement 1c and d*.

DOI: https://doi.org/10.7554/eLife.30688.011

The following figure supplement is available for figure 6:

**Figure supplement 1.** Mutagenesis used to identify the residues in the DBD that mediate the coupling to the R-domain.

DOI: https://doi.org/10.7554/eLife.30688.012

likelihood parameter estimation for $g_{R-D}$; Fig. S3e) and poor solubility of both the DBD and R-domains precluded attempts to structurally characterize the interaction using NMR, such limitations have not adversely affected our efforts to rationally intervene. Indeed, *Figure 6* reveals that the coupling between the R-domain and the DBD, identified in isolation (*Figure 3*), could not only be leveraged into a comprehensive frustration-based model that quantitatively captures the relative

binding and activity of all the GR isoforms (*Figure 5b*), but could also be rationally manipulated, and the opposing consequences on DNA binding and transcriptional activity predictably altered (*Figure 6e*). Thus, although physical basis of the couplings between the domains awaits future studies, the fact that GR has evolved the ability to produce isoforms that utilize different degrees of energetic frustration opens entirely new avenues for investigating regulation in IDPs.

The regulatory role of the isoform-specific ID R-domain in GR is especially important in light of the observation that the DNA sequences coding for ID regions are enriched in splice sites (*Buljan et al., 2012*), leading to a high degree of variability in the ID regions of the resultant proteins. The studies presented here provide a functional explanation. Alternative splicing, like the alternative translation start sites of GR described here, can produce proteins with different degrees of frustration in their ID regions, and thus differing activities. In addition, isoforms may also have different combinations of post-translational modification sites, which are also enriched in ID segments (*Bah and Forman-Kay, 2016*). By combining regulatory elements possessing different stabilities with different numbers and types of modification sites, ID proteins can potentially regulate not only the efficiency of the resultant protein, as shown here for GR, but also how that activity can be tuned by different types of modifications (*Motlagh et al., 2014*).

## Conclusions

We have shown that GR produces different isoforms, which have different DNA-binding affinities and transcriptional activities that are uncorrelated to each other. Our results show that this uncorrelated behavior is facilitated through 'energetic frustration', wherein opposing energetic couplings compete to modulate the overall response. Recent studies reveal that in addition to being facilitated by structured proteins, allostery can also be mediated by dynamic and even ID proteins (*Freiburger et al., 2011*; *Motlagh et al., 2014*; *Petit et al., 2009*; *Popovych et al., 2006*; *Tzeng and Kalodimos, 2009*, *2012*). Within these ubiquitous ID systems, significant heterogeneity, both in the apo and ligand-bound states, produces ensembles that cannot be treated using classic deterministic or structure-based allosteric models. Instead, extension of these classic models to account for positive and negative couplings between different regions provides a framework for understanding not only how ID sequences communicate with other structured and ID sequences, but also how such heterogeneity can produce complex regulatory strategies, such as the frustration-based mechanism identified here.

## Materials and methods

### Plasmids

DNA 2.0 (Menlo Park, CA) synthesized the plasmid used to express the A isoform of human GR in U-2 OS cells. The construct sequence was codon optimized and inserted into the PJ603 mammalian expression vector under CMV promoter control. Plasmids to express isoforms B, C1, C2, C3, D1, D2, D3 and DBD were made by inserting the codons for each respective isoform amplified from A isoform vector into the NheI and XhoI sites of the PJ603 vector. The GR F-Gal4 DBD plasmid was also produced by DNA 2.0 using the PJ603 plasmid backbone. The $R_A$-linker-DBD (equivalent to $R_A$-11aa linker-DBD)/$R_A$-20aa linker-DBD plasmid was made using a PCR that deleted the codons for GR 98–420 from the GR A isoform plasmid, then digesting with BamHI and KpnI, and ligating the sticky ends to an oligo coding for the 11aa linker GTGGSGGSGGS/20 aa linker GTGGSGGSGGSGGSGG SGGS. Plasmids for $R_A\Delta$86–97-linker-DBD, $R_A\Delta$27–97-linker-DBD, $R_B$ -linker-DBD, $R_{C1}$ -linker -DBD, $R_{C2}$ -linker and linker-DBD were made by inserting the codons for each construct amplified from $R_A$-11aa linker-DBD plasmid into the NheI and XhoI site of PJ603 vector. For the R domain-nuclear localization sequence-Flag construct, a GeneBlock was synthesized by IDT (Coralville, IA) to contain the GR R-region (1–97 a.a.), a four amino acid GSGS linker, the GR nuclear localization sequence (488– 505 a.a.), a GSGSGS linker, and the FLAG tag (DYKDDDDK). This GeneBlock was restriction digested with NheI and XhoI, then inserted into the pJ603 vector (DNA2.0). The FLAG tag was used for immunostaining to verify nuclear localization (data not shown). All the point mutations on GR constructs were made by site directed mutagenesis (*Hemsley et al., 1989*).

To measure transcriptional activity in the dual luciferase reporter assay, two tandem full length GREs (5'-aattcAGAACAggaTGTTCTgagatccgtagcAGAACAggaTGTTCTgagatccgtagcg −3') were

cloned into the EcoRI and BamHI sites in the promoter region of pGluc-miniTK vector (NEB, Ipswitch, MA), which expresses a secreted *Gaussia* luciferase (Tannous et al., 2005). For the competitive transfection assay, four tandem half-site GREs (5'-aattcAGAACAggagagatcgtagc AGAACAggaagatccgtagcAGAACAggagagatccgtagcAGAACAggaagatccgtagcg-3') were cloned into the promoter region of pGluc-miniTK vector. The pCluc-miniTK2 vector (NEB), expressing a secreted *Cypridina* luciferase (Nakajima et al., 2004) independent of GR regulation, was utilized as an internal control in the transfection to account for differences in cell density and transfection efficiency in each well.

DNA 2.0 also synthesized the plasmid for bacterial expression of the two-domain GR construct. Codons for the two-domain construct of A isoform was optimized for bacterial cell expression and inserted into the PJ411 expression vector under T7 promoter control. Plasmids to express isoforms B, C1, C2, C3, D1, D2 and D3 in *E.coli* were made by inserting the codons for each respective isoform amplified from A isoform plasmid into the NdeI and XhoI sites of the PJ411 vector.

## Protein expression and purification

Expression, purification and storage of the two-domain constructs for the eight GR translational isoforms were the same as for the single N-terminal domain construct, as described previously (Li et al., 2012), except for the following modifications in the purification steps. The lysis buffer was composed of 100 mM NaH$_2$PO$_4$, 10 mM Tris, 500 mM NaCl, 20 mM imidazole, pH 8.0. The wash buffer was the lysis buffer containing 60 mM imidazole, and the elution buffer was the lysis buffer containing 200 mM imidazole.

## DNA-binding affinity monitored by fluorescence anisotropy change

Fluorescent Oligos containing half site GRE (5'−6-FAM gcgcAGAACAggacgcg-3' and 5'-cgcgtccTGTTCTgcgc-3') were synthesized by IDT with HPLC grade purification and annealed with each other to get double stranded 6FAM-labeled half site GRE. The binding experiments of GR two domain constructs with the half site GRE were carried out in the following buffer: 10 mM HEPES (pH7.6), 80 mM NaCl, 1 mM EDTA, 5 mM MgCl$_2$, 1 mM DTT, 200 ug/mL BSA and 5 µM double strand control oligo (5'-GCGCCATATGATACGCG-3'). For each data point, 25 nM 6-FAM-labeled half site GRE was incubated with from 0 µM to 10 µM GR two-domain construct at 22°C for 30 min. Fluorescence anisotropy was measured using an Aviv ATF 105 fluorometer equipped with polarizers. A 'sub micro' fluorometer cell with 150 µL of solution was allowed to rest at 22°C (Santa Cells) for 2 min to allow for temperature stabilization and then excited at 495 nm. Anisotropy at 521 nm was recorded as a function of GR construct concentration and fitted with a single-site-binding model.

## Osmolyte TMAO induced protein folding transitions

TMAO-induced protein folding transitions were described previously (Li et al., 2012).

## Cell culture

U-2 OS cells (American Type Culture Collection, Manassas, VA) were maintained in modified McCoy's 5a medium (Corning Cellgro, Tewksbury, MA) supplemented with 10% fetal bovine serum and 100 U/mL penicillin and 100 µg/mL streptomycin. To transfect U-2 OS cells at about 80–90% confluence, X-tremeGENE HP DNA transfection reagent (Roche, Indianapolis, IN) was used at 2 µl per 1 µg DNA according to the manufacturer's manual.

## Transcriptional activity by dual luciferase reporter assay

For the transcriptional activity dosage curve, 40 ng of pGluc-miniTK vector with two tandem full length GREs cloned in the promoter region, 40 ng of pCluc-miniTK2 and up to 5 ng (saturating) of GR expression vector were co-transfected into U-2 OS cells on 96-well plates. For the competitive transfection assay, 40 ng of pGluc-miniTK vector with four tandem half site GREs cloned in the promoter region, 40 ng of pCluc-miniTK2, 3 ng of expression vector for C3 isoform, and up to 16 ng of plasmid coding for one of the competitors were co-transfected into U-2 OS cells on 96-well plates. For titration of the C3 isoform wild type and C3 C431Y&V435A&L436A mutant with the R domain-nuclear localization sequence-Flag construct, the method was the same as the competitive transfection assay described above, except up to 12 ng of the R domain-nuclear localization sequence-Flag

construct plasmid was used in the titration. After 48 hr, *Gaussia* Luciferase activity and *Cypridina* Luciferase activity were measured with the BioLux *Gaussia* Luciferase Assay Kit (NEB) and the BioLux *Cypridina* Luciferase Assay Kit (NEB), respectively, on a TriStar LB 942 Multidetection Microplate Reader (Berthold Technologies GmbH & Co. KG, Bad Wildbad, Germany), according to the manufacturer's protocols. In each experiment, the *Gaussia* luciferase activity (normalized by the *Cypridina* luciferase activity) was measured in triplicate and averaged. The dosage and competitive transfection curves were fitted with dose response curves using Origin.

## Western blot

U-2 OS cells were plated on 6-well plates at a density of $5 \times 10^5$ cells per well. After 18 hr, 50 ng of GR expression vector and 450 ng of salmon sperm DNA (Invitrogen, Carlsbad, CA, transfection boosting reagent) were transfected into each well with X-tremeGENE HP DNA transfection reagent (Roche), following the manufacturer's protocol. The medium was changed once 24 hr post transfection. After 48 hr, the cells were scraped from each well with PBS, and pelleted by centrifuging at 1500 rpm. For lysis, 50 µL of lysis buffer (8 M urea, 20 mM Tris-HCl, 500 mM NaCl, 1 mM $Na_2EDTA$, 1 mM EGTA, 1% Triton, 2.5 mM sodium pyrophosphate, 1 mM beta-glycerophosphate, 1 mM $Na_3VO_4$ and 1 µg/ml leupeptin, pH 7.5) was added to each cell pellet. To reduce the viscosity, cells were passed through a 26G 3/8'' syringe needle 10 times and then the cell lysate was centrifuged at 14000 rpm for 30 min. Supernatant was collected and the total protein concentration was measured by Bradford assay (Bio-Rad, Hercules, CA). In each well of a 4–15% Mini-PROTEAN TGX Precast Gel (Bio-Rad), 5 µg of total protein was loaded, and separated in Tris/Glycine/SDS gel running buffer. Transfer of the protein from the SDS page gel to PVDF film was done in the transfer buffer (25 mM Tris-HCl, pH 8.3, 192 mM glycine, 20% methanol) under 120V for 15 min. After blocking in 5% nonfat milk in PBS with 0.1% Tween-20 (PBST) for 1 hr, the PVDF film was then incubated at 4°C overnight with 10000 fold diluted primary antibody for GR (BD Transduction Laboratories, #611226, San Jose, CA) or p150glued (BD Transduction Laboratories, #610473), which served as loading control. Both antibodies were diluted into the same 5% nonfat milk in PBST. The next morning, after washing with PBST buffer for three times, the PVDF film was incubated in the 20000 fold diluted HRP-linked anti-mouse IgG (GE healthcare, NA931, Chicago, IL), also in the 5% nonfat milk in PBST. The detection was done with Amersham ECL Prime Western blotting reagent (GE heathcare, RPN2232) and autoradiography film (Denville Scientific).

## Immunostaining

U-2 OS cells were plated on 6-well plates with 15 mm round German coverslips. All the culture and transfection procedures are the same as done for the cells for western blot experiments. After 48 hr, cells were rinsed with PBSM (PBS with 2 mM $MgCl_2$) three times, and fixed with 4% paraformaldehyde in PBSM at room temperature for 10 min. Afterwards, each coverslip was rinsed with PBSM three times again, and quenched with 50 mM $NH_4Cl$ in PBSM. Then the slide was placed in PBSTB (PBS with 0.1% Triton X-100, 1% BSA) for 30 min at room temperature to permeabilize cells and block nonspecific binding. Thereafter, the slide was incubated for 1 hr at room temperature with primary rabbit antibody for GR (cell signaling, #3660), which was 5000 fold diluted in PBSTB. Then the slide was washed three times with PBSM and incubated for 30 min at room temperature in the dark with the Alexa Fluor 488 Goat Anti-Rabbit IgG (Invitrogen, Carlsbad, CA), which was 600 fold diluted in PBSTB. Next the slide was incubated for 10 min at room temperature in the dark in PBSM with 0.2 µg/mL DAPI (Invitrogen) and 5 unit/mL Rhodamine Phalloidin (Invitrogen) to stain the nuclei and F-actin respectively. After that, each slide was washed with PBSM twice and mounted onto a microscope slide with Fluoromount (Sigma, St. Louis, MO), and kept in the dark for drying.

Images were taken with an inverted light microscope (Axiovert 200, JHU Integrated Imaging Center). All images were taken the same day using the same gain, exposure times, and filter configurations (DAPI, FITC, and Texas Red filters). The images were analyzed using ImageJ (*Staal et al., 2004*).

## Ensemble allosteric model

The Ensemble Allosteric Model (EAM) and its usage have been described in detail previously (*Hilser and Thompson, 2007*; *Hilser et al., 2012*; *Motlagh et al., 2014*).

## Acknowledgements

This work was supported by National Science Foundation grant MCB1330211 and National Institutes of Health grants GM-063747 and T32-GM008403.

## Additional information

### Funding

| Funder | Grant reference number | Author |
|---|---|---|
| National Science Foundation | MCB-1330211 | Jing Li<br>Jordan T White<br>Harry Saavedra<br>James O Wrabl<br>Hesam N Motlagh<br>Kaixian Liu<br>James Sowers<br>Vincent J Hilser |
| National Institutes of Health | GM-063747 | Hesam N Motlagh<br>James O Wrabl<br>Vincent J Hilser |
| National Institutes of Health | T32-GM008403 | Hesam N Motlagh<br>James O Wrabl<br>Vincent J Hilser |
| Johns Hopkins University | JHU Institutional Funds | Vincent J Hilser |

The funders had no role in study design, data collection and interpretation, or the decision to submit the work for publication.

### Author contributions

Jing Li, Conceptualization, Formal analysis, Investigation, Visualization, Methodology, Writing—original draft, Writing—review and editing; Jordan T White, Formal analysis, Validation, Investigation, Visualization, Methodology, Writing—review and editing; Harry Saavedra, Validation, Investigation, Methodology, Writing—review and editing; James O Wrabl, Software, Formal analysis, Visualization, Methodology, Writing—review and editing; Hesam N Motlagh, Software, Formal analysis, Investigation, Methodology, Writing—review and editing; Kaixian Liu, James Sowers, Investigation, Methodology; Trina A Schroer, Resources, Formal analysis, Supervision, Validation, Project administration, Writing—review and editing; E Brad Thompson, Conceptualization, Formal analysis, Supervision, Investigation, Project administration, Writing—review and editing; Vincent J Hilser, Conceptualization, Supervision, Funding acquisition, Validation, Visualization, Writing—original draft, Project administration, Writing—review and editing

### Author ORCIDs

Jordan T White http://orcid.org/0000-0003-3202-4181
Trina A Schroer https://orcid.org/0000-0002-5065-1835
E Brad Thompson http://orcid.org/0000-0003-1578-0241
Vincent J Hilser http://orcid.org/0000-0002-7173-0073

### Decision letter and Author response

Decision letter https://doi.org/10.7554/eLife.30688.016
Author response https://doi.org/10.7554/eLife.30688.017

## Additional files

### Supplementary files

• Supplementary file 1. Mathematica notebook for data fitting.
DOI: https://doi.org/10.7554/eLife.30688.013

• Transparent reporting form

DOI: https://doi.org/10.7554/eLife.30688.014

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
