## [Decision Letter]

Thank you for submitting your article "Genetically tunable frustration controls allostery in an intrinsically disordered transcription factor" for consideration by *eLife*. Your article has been very favorably reviewed by three peer reviewers. Indeed, it relatively rare for us to see such consistency in praise for a paper under evaluation.

The evaluation has been overseen by John Kuriyan as the Reviewing Editor. The following individuals involved in review of your submission have agreed to reveal their identity: Rohit V Pappu (Reviewer #1); David Eliezer (Reviewer #2).

The manuscript needs revision, but as you will see, we are only asking for textual changes. None of the issues raised are considered to be particularly serious, and you should use your best judgement as to how to address them.

Overall comments:

Reviewer 1

This work by Hilser and coworkers continues their efforts to uncover the role of intrinsically disordered regions in allostery as exemplified in the GR family of hormone regulated transcription factors. This MS is well written, clear, and easy to follow despite the numerous constructs that one has to keep in mind. The major theme in this work, which is a continuation of the ensemble allostery model for TFs with IDRs, is the finding that different GR isoforms, distinguished entirely by differences in their N-terminal IDRs, engender different transcriptional activities entirely through differences in coupling free energies among different domains. This result shines through because the DBDs are identical in all of the constructs, and yet the authors observe a clear impact of changes to the N-terminal IDR on DNA binding and on transcriptional activity, although there is no correlation between the DNA binding affinities and activities. This family of TFs is an elegant system for dissecting the impact of IDR mediated allostery and the model appears to do a good job of recapitulating the measured activities.

Overall, this is a very strong paper, clearly written, espousing coherent concepts, and highlighting novel roles for IDRs, which go beyond speculation and anchor the findings in concrete data, that are driven by predictions. In many ways, this work exemplifies the best of science and the integrative efforts of clearly formulated statistical physics, biophysical experimentation, and evolutionary principles.

Reviewer 2

This paper describes a study to document the previously predicted phenomenon of allostery involving disordered protein regions using the glucocorticoid receptor (GR) as a model system. GR contains two functionally separable disordered regions, each of which is shown experimentally to interact favorably with the DNA-binding domain (DBD), but to interact unfavorably with each other, resulting in a frustrated system in which activity, mediated by the F domain, is enhanced directly by the DBD but is repressed indirectly via DBD-enhanced repression by the R domain. Using the model previously developed to model allostery in such system, featuring the individual stabilities of the domains and the coupling energies between each pair of domains, the authors are able to generate results that are in delightful agreement with the experimental measurements on differently spliced isoforms of GR in which the R region is present to different degrees. Even more impressively, a series of mutations is identified which are able to disrupt the coupling between the DBD and the R domain, which then show precisely the predicted effects on activity, both in cis and in trans.

This is a lovely, clear paper that beautifully demonstrates the validity and biological relevance of the model previously developed by the authors for allostery involving disordered protein regions. This is a conceptually important extension of the classical models of allostery, and in light of the ever growing importance of disordered protein regions, is a critical contribution to our understanding of protein function regulation.

Reviewer 3

In this study by Li et al. the authors have investigated the role of regions of the intrinsically disordered amino-terminal domain of the glucocorticoid receptor in allosteric regulation. Specially they have tested experimentally and further validated their previously published 'ensemble allosteric model'.

This is an excellent study and significantly contributes to our growing understanding of the role of intrinsically disordered structure and nuclear receptor signalling. The authors have used a range of molecular and biophysical methods to investigate receptor-dependent transcription, DNA binding affinity and protein folding.

Comments to address:

1) Could the title be made somewhat more accessible? The average *eLife* reader may not be "tuned" to the concept of "frustration", and there is some concern that the title may not draw in all readers who should be interested in this work.

2) The terminology regarding frustration should be clarified and improved. First, several declarations are made about the history of frustration that are inaccurate. The triangular Ising net was introduced by Wannier in 1950. The discovery of frustration in ice was explained by Pauling in 1935. And the term "energetic frustration" is a concoction. In condensed matter physics, the term is "geometric frustration" or just "frustration". There is a lot more to frustration theory than is alluded to here. It would be helpful to have a paragraph laying out the frustration models that the authors are connecting to, explain why this is a useful concept, and map back to this model with their data. Otherwise, there is concern that *eLife* readers may not find the analogies useful.

3) In the context of comment 2) above, it would help to clarify how the classification into frustrated vs. non-frustrated configurations was achieved in Figure 4. It looks like there are configurations that are erroneously labeled as non-frustrated ones and this is why the precepts of the model would be useful to clarify.

4) The paper does not clearly explain how the model is able to differentiate between isoforms that feature different length deletions of the R domain (isoforms A, B, C1, C2 and C3). As far as I can tell, the model only accounts for the presence of the R domain (isoform A) or its absence (isoform C3), yet predictions and experimental results are shown (Figure 5) for isoforms C1, C2 and C3 as well. Clarification of how this was accomplished would be welcome.

5) Many of the reported errors are noted to represent the uncertainty of the curve fits to the date, but details regarding how the curve fitting was performed and how the errors were then extracted would be welcome in the supplementary materials.

6) Regarding the choice of response elements in the different transcription and DNA binding experiments: The authors have used a standard reporter gene containing palindromic glucocorticoid response elements (GRE). However, in the transcription competition studies and DNA binding studies the focus appears to have been on half-sites (four or single respectively). The rationale for this is not clear and the binding affinity to a half site is almost certainly different from a 15 bp GRE. This needs to be made clear and the rationale for using half-sites explained and the broader consequences of the results to glucocorticoid receptor transcription. Did the authors wish to avoid dimerization of receptor isoforms complicating the measurements? Please clarify.

7) It would have been nice to see data for bona fide GREs from known glucocorticoid target genes, as these a likely to exhibit different binding affinities and/or structures, but of course there has to be a balance with technical consideration. However, the authors could comment on the action of the glucocorticoid receptor on negative GREs (ie Hua et al., 2016) with regards their allosteric model (Figure 4). The 'non-frustrated' configurations would appear to be more reflective of known glucocorticoid receptor action?

8) Figure 6. Could the data for C3 and DNA binding domain mutant C3 be shown in part 'E'? The prediction would be that C3 activity would be immune to the triple mutation. This seems to be shown in part 'f' 0 sample, but it would be nice to see it clearly shown as it provides further support for the author's model.

---

## [Author Response]

Comments to address:1) Could the title be made somewhat more accessible? The average eLife reader may not be "tuned" to the concept of "frustration", and there is some concern that the title may not draw in all readers who should be interested in this work.

We agree with the reviewer’s concern that the concept of “frustration” is not widely used in the biological sciences, but as it is the main point we belief it will actually draw readers. In short, we believe the title will turn out to be a strength, since the unusual terminology, for the average *eLife* reader, could spark curiosity and debate across disciplines.

In support of this choice, a recent high-profile review states that “… Frustration is a fundamental concept in molecular biology” (Ferreiro, et al., 2014). Our title may thus be viewed as raising current awareness of a concept that has latently existed in the biological literature for decades. We respectfully opt to retain the title without modification.

2) The terminology regarding frustration should be clarified and improved. First, several declarations are made about the history of frustration that are inaccurate. The triangular Ising net was introduced by Wannier in 1950. The discovery of frustration in ice was explained by Pauling in 1935. And the term "energetic frustration" is a concoction. In condensed matter physics, the term is "geometric frustration" or just "frustration". There is a lot more to frustration theory than is alluded to here. It would be helpful to have a paragraph laying out the frustration models that the authors are connecting to, explain why this is a useful concept, and map back to this model with their data. Otherwise, there is concern that eLife readers may not find the analogies useful.

We thank the reviewer for pointing out how this manuscript could be more effectively couched in the historical literature.

We agree with the reviewer that the term “energetic frustration”, undeveloped as it was in the previous version, was a concoction. Because of the more than superficial analogy with the physics models, it deserves a distinguishing name. For consistency, the term “energetic frustration” has been adopted throughout the text.

Furthermore, we agree with the reviewer that our terminology regarding energetic frustration could have been articulated more clearly. Accordingly, we clarify our model in subsection “The Ensemble Allosteric Model (EAM) Enables Quantitative Characterization of the Energetic Frustration in GR”, and its analogy to geometric frustration, as follows. Our model of energetic frustration in GR is related, but not identical, to classical physics models for the interaction energies between pairs of spins in a magnet. The simplest model of spin interaction energy takes the form E_int_ = -J_ij_S_i_S_j_, where J_ij_ is the coupling between spins i and j, while S_i_, S_j_ are the signs referring to up/down nature of the spins (Ferreiro, et al., 2014). Similarly, we model the interaction energy between one pair of domains in GR (or any multi-domain protein) with an analogous mathematical form, as E_int_ = J_ij_(S_i_|∆G_i_)(S_j_|∆G_j_). In our model, J_ij_ is also a coupling energy, which we instead name ∆g_int_, between the domains.

There are, however, three key differences between the models. First, S_i_ and S_j_, though still signs as in the classical model, are constrained to the values of {0, +1}. Second, the value of the sign is conditional on the stabilities of the individual domains, ∆G_i_ and ∆G_j_. If the domains are both folded, the interface between them is also folded. S_i_ and S_j_ then both take the value +1, and only in this case does the complete expression provide additional coupling energy ∆g_int_ to the system. Third, the sign of J_ij_ in our model may be either positive or negative, as allosteric coupling in different proteins has been observed to be either positive or negative.

These differences increase the theoretical complexity of the energetic frustration model, relative to the simple geometric frustration model. Thus, connecting our Ensemble Allostery Model (EAM) with physics frustration models is a useful analogy because the non-intuitive allosteric behavior exhibited by biological systems, such as GR, may be now partially understood and interpreted in terms of the classical physics models, which have been more thoroughly explored for almost a century.

3) In the context of comment 2) above, it would help to clarify how the classification into frustrated vs. non-frustrated configurations was achieved in Figure 4. It looks like there are configurations that are erroneously labeled as non-frustrated ones and this is why the precepts of the model would be useful to clarify.

We are grateful to the reviewer for making this point, as it highlights the shortcomings in the original version in explaining our model and differentiating energetic frustration from geometric frustration. We have clarified the description of Figure 4 by stating that all configurations in the Figure are correctly labeled with respect to energetic frustration. The reason for this is that the triplets of arrows in the Figure represent a different type of interaction than the spin pairs of geometric frustration, and thus individual arrows do not represent molecular positions and neighboring pairs of arrows should not be interpreted as directly interacting with each other. Rather, each triplet arrow in the context of energetic frustration represents a domain-domain interface, and the up/down direction of the arrow represents positive or negative allosteric coupling. In the particular case of GR, these arrows are interpreted from left to right as representing the coupling between DBD- and F-domains, the coupling between R- and F-domains, and the coupling between DBD- and R-domains, respectively. DNA binding to the DBD stabilizes the DBD, and thus inputs a positive coupling energy into the system. The response of the system, i.e. transcriptional regulation, is a complex function of the competition between the direct coupling between DBD-F and the indirect coupling mediated through the R-domain.

Put another way, the energetic frustration of the system results from the numbers of up and down arrows in the triplet without regard to the exact ordering: for example, two up arrows grouped with one down will always result in a system with energetic frustration, given an input of positive coupling. Although we thought adequate explanation was given in the legend, we see now that a clearer description was necessary. We sincerely apologize for all these shortcomings. To address the reviewer’s concerns and to improve the clarity of Figure 4 triplet has been added to 4A to better connect 4A to 4B, and the text and figure legend have been expanded.

4) The paper does not clearly explain how the model is able to differentiate between isoforms that feature different length deletions of the R domain (isoforms A, B, C1, C2 and C3). As far as I can tell, the model only accounts for the presence of the R domain (isoform A) or its absence (isoform C3), yet predictions and experimental results are shown (Figure 5) for isoforms C1, C2 and C3 as well. Clarification of how this was accomplished would be welcome.

All the detailed constraints for A, B, C1, C2, and C3 isoforms are in the Mathematica notebook. Indeed, the reviewer is correct that emphasis has been placed in the text on A and C3, even though the Ensemble Allosteric Model (EAM) implemented for GR can treat the other isoforms.

The key assumption influencing the predictions for the B, C1, C2 and C3 isoforms, relative to A, is a length-dependent coupling energy between DBD- and R-domains: since the A isoform of the R-domain is 97 residues and the B isoform is 71 residues we assume the coupling energy between B and DBD is roughly 80% of that observed between A and DBD. This modification is coded to variable “SWdbdRB” (abbreviation for “Statistical Weight of folded DBD and folded R domain, B-isoform”) in the fifth line of the “Partition Function for the B Isoform" in the Supplemental Mathematica Notebook. Similarly, every C isoform has zero R-domain and DBD coupling energy, since the entire R-domain is missing in these isoforms; these modifications are coded explicitly as zero in line 5 of each of the C isoform partition functions.

Since the coupling energy between F-domain and DBD is also identical among the C1, C2, C3 domains, their different predicted values must arise from the fact that the EAM naturally permits different values for the intrinsic stabilities of C1 and C2 in the solution (the measured value for C3 is one of the experimental constraints).

5) Many of the reported errors are noted to represent the uncertainty of the curve fits to the date, but details regarding how the curve fitting was performed and how the errors were then extracted would be welcome in the supplementary materials.

Curve fitting was performed using Mathematica’s NonLinearModelFit function with all-default parameters, which automatically provides error estimates at 95% confidence intervals. These automatic error estimates are reported in the Figures without further modification. The NonLinearModelFit function assumes that errors are independent and normally distributed, and the reported fit minimizes the sum of the squared errors. Complete details can be found in the Mathematica documentation, located at: http://reference.wolfram.com/language/tutorial/StatisticalModelAnalysis.html

To address the reviewer’s concern, to the Figure supplements legends, where NonLinearModelFit was employed, we have added the phrase “…as returned by the default settings of Mathematica’s NonLinearModelFit function.”

6) Regarding the choice of response elements in the different transcription and DNA binding experiments: The authors have used a standard reporter gene containing palindromic glucocorticoid response elements (GRE). However, in the transcription competition studies and DNA binding studies the focus appears to have been on half-sites (four or single respectively). The rationale for this is not clear and the binding affinity to a half site is almost certainly different from a 15 bp GRE. This needs to be made clear and the rationale for using half-sites explained and the broader consequences of the results to glucocorticoid receptor transcription. Did the authors wish to avoid dimerization of receptor isoforms complicating the measurements? Please clarify.

We agree with the reviewer’s statement that the binding affinity to a half-site is almost certainly different than to a 15 bp GRE. Indeed, the reviewer is correct that we wished to avoid the complication of dimerization by using the half-site, since dimerization is reported to be palindromic-GRE dependent. In contrast, the four tandem half-site GRE was chosen in the cell assay because of its increased signal-to-noise ratio.

7) It would have been nice to see data for bona fide GREs from known glucocorticoid target genes, as these a likely to exhibit different binding affinities and/or structures, but of course there has to be a balance with technical consideration. However, the authors could comment on the action of the glucocorticoid receptor on negative GREs (ie Hua et al. 2016) with regards their allosteric model (Figure 4). The 'non-frustrated' configurations would appear to be more reflective of known glucocorticoid receptor action?

We thank the reviewer for bringing this paper to our attention, as we are unfamiliar with this particular work. As detailed in this paper, binding of an inverted repeated negative GRE (IR nGRE) seems to require conformational change and covalent modification within the NTD (i.e. “unmasking” as described by Hua, et al.), mediated by LBD binding of a glucocorticoid. Although the LBD and covalent modifications are not modeled in our work, we can accept the point that the DBD could be structurally malleable. Thus, we could speculate that IR nGRE binding may induce a conformation of DBD that is different from the positive GRE binding in our model. How this conformation could be coupled to the R and F domains are intriguing topics for future study. We think the applicable coupling scenarios would be cases ii, iii, vi, or vii in Figure 4.

8) Figure 6. Could the data for C3 and DNA binding domain mutant C3 be shown in part 'E'? The prediction would be that C3 activity would be immune to the triple mutation. This seems to be shown in part 'f' 0 sample, but it would be nice to see it clearly shown as it provides further support for the author's model.

We are grateful to the reviewer for such a close reading of our technical data! The C3 activity is indeed immune to the triple mutation, and the reviewer is correct that it is shown in the 0 sample in panel 6F. Both activity and binding data for C3-DBD and its triple mutant are already shown together in Figure 6—figure supplement 1, panels E and F. Although we think it reasonable to do as suggested, we respectfully elect not to increase the size and complexity of panel 6E.